# Genetic Ablation and Pharmacological Blockade of Bradykinin B1 Receptor Unveiled a Detrimental Role for the Kinin System in Chagas Disease Cardiomyopathy

**DOI:** 10.3390/jcm12082888

**Published:** 2023-04-15

**Authors:** Ana Carolina Oliveira, Amanda Roberta Revoredo Vicentino, Daniele Andrade, Isabela Resende Pereira, Leonardo Saboia-Vahia, Otacílio da Cruz Moreira, Carla Eponina Carvalho-Pinto, Julia Barbalho da Mota, Leonardo Maciel, Glaucia Vilar-Pereira, João B. Pesquero, Joseli Lannes-Vieira, Pierre Sirois, Antônio Carlos Campos de Carvalho, Julio Scharfstein

**Affiliations:** 1Programa de Imunobiologia, Instituto de Biofísica Carlos Chagas Filho, Universidade Federal do Rio de Janeiro, Rio de Janeiro 21941-902, Brazil; 2Laboratório de Biologia das Interações, Instituto Oswaldo Cruz, Fundação Oswaldo Cruz, Rio de Janeiro 21040-360, Brazil; 3Plataforma de PCR em Tempo Real RPT09A, Laboratório de Virologia Molecular, Instituto Oswaldo Cruz, Fundação Oswaldo Cruz, Rio de Janeiro 21040-360, Brazil; 4Departamento de Imunobiologia, Instituto de Biologia, Universidade Federal Fluminense, Niterói 24020-141, Brazil; 5Programa de Medicina Regenerativa, Instituto de Biofísica Carlos Chagas Filho, Universidade Federal do Rio de Janeiro, Rio de Janeiro 21941-902, Brazil; 6Núcleo Multidisciplinar de Pesquisa em Biologia, Universidade Federal do Rio de Janeiro, Duque de Caxias Campus, Rio de Janeiro 21941-902, Brazil; 7Departamento de Biofísica, Universidade Federal de São Paulo, São Paulo 05508-090, Brazil; 8Department of Microbiology and Immunology, Faculty of Medicine, Université Laval, Quebec, QC G1V 0A6, Canada; 9Centro Nacional de Biologia Estrutural e Bio-Imagem, Universidade Federal do Rio de Janeiro, Rio de Janeiro 21941-902, Brazil; 10Instituto Nacional de Ciência e Tecnologia em Medicina Regenerativa, Rio de Janeiro 21941-902, Brazil

**Keywords:** bradykinin, Chagas disease, GPCRs, kallikrein, *Trypanosoma cruzi*

## Abstract

Chagas disease, the parasitic infection caused by *Trypanosoma cruzi*, afflicts about 6 million people in Latin America. Here, we investigated the hypothesis that *T. cruzi* may fuel heart parasitism by activating B1R, a G protein-coupled (brady) kinin receptor whose expression is upregulated in inflamed tissues. Studies in WT and B1R^−/−^ mice showed that *T. cruzi* DNA levels (15 days post infection—dpi) were sharply reduced in the transgenic heart. FACS analysis revealed that frequencies of proinflammatory neutrophils and monocytes were diminished in B1R^−/−^ hearts whereas CK-MB activity (60 dpi) was exclusively detected in B1R^+/+^ sera. Since chronic myocarditis and heart fibrosis (90 dpi) were markedly attenuated in the transgenic mice, we sought to determine whether a pharmacological blockade of the des-Arg^9^-bradykinin (DABK)/B1R pathway might alleviate chagasic cardiomyopathy. Using C57BL/6 mice acutely infected by a myotropic *T. cruzi* strain (Colombian), we found that daily treatment (15–60 dpi) with R-954 (B1R antagonist) reduced heart parasitism and blunted cardiac injury. Extending R-954 treatment to the chronic phase (120–160 dpi), we verified that B1R targeting (i) decreased mortality indexes, (ii) mitigated chronic myocarditis, and (iii) ameliorated heart conduction disturbances. Collectively, our study suggests that a pharmacological blockade of the proinflammatory KKS/DABK/B1R pathway is cardioprotective in acute and chronic Chagas disease.

## 1. Introduction

Transmitted to humans by hematophagous triatomine insects, the hemoflagellate protozoan *Trypanosoma cruzi* is the etiologic agent of Chagas disease (CD), a chronic parasitic disease that affects approximately 6 million people in Latin America [1]. The classical mode of *T. cruzi* transmission to mammals involves deposition of the metacyclic trypomastigotes (insect-derived infective forms) on tissues lacerated by an insect’s proboscis. However, accidental cases of oral infection have increased due to ingestion of juices or food stocks contaminated with insect-derived parasites [2]. Alternative transmission of CD involves congenital infection, blood transfusion, or organ transplantation. Owing to immigration, asymptomatic individuals have gradually spread CD to developed countries [3]. Following a long period of clinically silent infection, a progressive chronic cardiomyopathy develops in approximately 30% of the chagasic patients. Associated with chronic myocarditis and fibrosis, chronic chagasic cardiomyopathy (CCC) may lead to congestive heart failure [4]. A lower proportion of chronic chagasic patients are afflicted by digestive mega syndromes, mostly affecting the esophagus/colon. Unfortunately, the progression of CCC was not halted by currently available antiparasitic drugs [4] nor by cell therapy with autologous bone marrow-derived cells [5]. To prevent fatal outcomes in a limited number of CCC patients, heart transplantation is occasionally performed in specialized hospitals [4].

*T. cruzi* is a genetically diversified species of parasitic protozoa [6] whose life cycle in mammals require obligate cycles of intracellular replication and morphogenetic transformation. At the end of each intracellular cycle of parasite development, a large number of flagellated parasites (trypomastigotes) are released to the interstitial spaces. Empowered with an impressive array of virulence factors, including highly polymorphic surface glycoproteins [7], the extracellular trypomastigotes invade macrophages as well as a broad range of non-professional phagocytic host cells (e.g., smooth muscle, cardiomyocytes, endothelial cells, fibroblasts and epithelial cells). After escaping from the parasitophorous vacuole, *T. cruzi* transforms into amastigotes in the host cell cytoplasm. Following several rounds of binary division, a large number of pear-shaped amastigotes transform into flagellated (infective) trypomastigotes. The asynchronous cycle of *T. cruzi* development [8,9], which usually lasts about 5–7 days in tissues, terminates when the intracellular trypomastigotes egress from dying host cells. After falling in the bloodstream, the flagellated trypomastigotes disseminate systemically, preferentially infecting macrophages in the spleen and lymph nodes. Patients with acute CD display a high blood parasitemia which gradually subsides with the onset of adaptative immunity. A long period of clinically silent infection is established in which host/parasite equilibrium is maintained. Whether sheltered in human adipose tissues or other immunoprivileged sites [10,11], low numbers of *T. cruzi* escape from cytotoxic effector CD8^+^ T cells [10]. Typically acting as an opportunistic pathogen, *T. cruzi* exploits oxidative stress to enhance intracellular outgrowth in macrophages [11]. Another example of the biological versatility of *T. cruzi* emerged from intravital microscopy studies showing that infection-driven microvascular leakage in the hamster cheek pouch depends on the functional interplay between mast cells and the kallikrein–kinin system (KKS) [9,12]. The groundwork laid by these mechanistic studies led us to demonstrate that bradykinin (BK)-induced inflammatory cascades provide the trypomastigotes with a window of opportunity to parasitize heart tissues of mice via cross-talk between G protein-coupled bradykinin B2 receptors (B2R) and endothelin receptors [12,13,14]. 

Faced with the challenge to investigate the pathogenic mechanisms underlying CCC, Teixeira et al. have recently reported proteomic data obtained with myocardium samples from CCC patients. Their molecular studies suggest that the patients’ susceptibility to CCC is increased due to dysfunctions in mitochondrial energy metabolism [15]. Whether caused by persisting or intermittent cycles of heart parasitism [16], the retention of minute quantities of *T. cruzi* antigens in the myocardium is sufficient to activate heart-infiltrating effector T cells. Using mice chronically infected with a myotropic *T. cruzi* strain (Colombian), Silverio et al. have linked the severity of CCC to unbalanced intracardiac infiltration of pathogenic subsets (perforin-expressing) of *T. cruzi*-specific CD8^+^ T cells [17]. Immune profiling of the peripheral blood of patients at different clinical stages of CD suggested that the clinical outcome of CCC might be worsened due to immunoregulatory dysfunctions [18]. 

Several years ago, proponents of the vascular theory [19,20] suggested that infection-driven microvasculopathy and T cell-dependent immunopathology might converge, aggravating the outcome of CCC. Intriguingly, histopathological analyses of *postmortem* heart sections revealed the presence of enlarged arterioles and tortuous capillaries in the left ventricle of CCC [21]. Since histopathological analysis of heart sections from CCC patients are inevitably limited to autopsies or biopsies from patients that underwent heart transplantation, it is still unknown whether the low-grade parasitism causes damage to cardiac capillaries during the transition from acute to chronic stage of infection. In a study of heart microangiopathy in *T. cruzi*-infected dogs, Andrade et al. described the presence of damaged capillaries containing immune cells associated to fibrin microthrombi [22]. Pioneer studies performed by Tanowitz et al. demonstrated that heart fibrosis is worsened by endothelin-1, a vasoconstrictor and profibrotic polypeptide whose expression is upregulated in parasitized heart cells [23]. In parallel, systematic analyses of the role of the KKS cascade in dynamics of *T. cruzi*-elicited inflammation revealed a dual role for B2 receptors. On one hand, trypomastigotes increment their infectivity at the expense of mast cell-driven activation of the KKS/B2R pathway [13,14]. Reciprocally, the short-lived BK stimulates the maturation of B2R-expressing dendritic cells, which in turn coordinate the intralymphoid development of immunoprotective type-1 effector CD4^+^ and CD8^+^ T cells via the IL-12 pathway [24,25]. 

Tightly regulated by C1 inhibitor and various metallopeptidases, the KKS is a proteolytic hub that interconnects proinflammatory and procoagulant cascades to innate/adaptive immunity [26]. Plasma kallikrein (PK) and tissue kallikrein, respectively, liberate the nanopeptide BK or lysyl-BK (LBK) from an internal domain of high and low molecular weight kininogens (HK/LK) [27]. During *T. cruzi* infection, the major parasite cysteine protease (cruzipain) promotes the release of bioactive kinins from kininogen molecules that are bound to heparan sulfate proteoglycans [28]. Further downstream, BK-induced inflammatory cascades are propagated via iterative cycles of mast cell degranulation and extravascular activation of plasma-borne contact (PK, FXII) factors [14].

Kinins exert their vascular functions, such as nitric oxide (NO) dependent vasodilation, microvascular leakage and angiogenesis by signaling two different subtypes of G protein-coupled receptors (GPCRs), kinin B1 (B1R), and B2R [29]. Activated by primary kinins (BK/LBK), B2R is constitutively expressed by a variety of cell types, including vascular endothelial cells, cardiomyocytes, smooth muscle cells, nociceptive neurons, macrophages, and dendritic cells [24,26,30,31]. In contrast, B1R is poorly expressed in most tissues [32], except for adipocytes [33]. However, in injured/inflamed tissues, the transcriptional expression of B1R is vigorously stimulated via the NF-kB pathway by several proinflammatory cytokines, such as IL-1β and TNF-α [34]. Unlike B2R agonists, B1R is triggered by des-Arg^9^-BK (DABK) or by lysine-des-Arg^9^-BK (LDABK). In both cases, the generation of B1R agonists requires the proteolytic excision of the C-terminal Arginine of primary kinins (BK/LBK) by two alternative forms of kininase I: a GPI-linked membrane-bound carboxypeptidase M or soluble carboxypeptidase N [32,35]. 

Long-range effects of BK/LBK on B2R are efficiently constrained by kinin-degrading peptidases, such as angiotensin-converting enzyme (ACE) [36]. Notably, there are striking differences in the mechanisms regulating B2R versus B1R signaling responses. While the constitutively expressed B2R is rapidly downmodulated upon agonist binding, the ligation of DABK to B1R stabilizes its endothelial surface expression [37]. Importantly, the levels of DABK are controlled by ACE2, a surface metallopeptidase that plays a dual anti-inflammatory role: ACE2 (i) degrades DABK (B1R agonist) and (ii) modulates the renin-angiotensin system (RAS) by converting angiotensin II, a potent hypertensive and profibrotic peptide, into the vasodilator/cardioprotective angiotensin 1–7 [38,39,40]. Interestingly, ADAM-7 (a disintegrin and metallopeptidase domain 17), also known as TNF-α converting enzyme (TACE), is a type-1 trans-membrane metallopeptidase that catalyzes ACE2 (ectodomain) cleavage from cell surfaces [40]. 

In the last decades, systematic studies of the role of B2R and B1R in mice models of myocardial ischemia and experimentally induced diabetic cardiomyopathy converged at appointing cardioprotective roles for the KKS [41,42,43]. In a seemingly contradictory finding, however, Westermann et al. reported that diabetic cardiomyopathy induced by streptozotocin was ameliorated in mice genetically deficient of B1R [44]. In the current work, we provide evidence that the genetic deficiency and pharmacological blockade of B1R reduces heart parasitism and ameliorates chronic heart pathology in mice models of Chagas disease. 

## 2. Material and Methods 

### 2.1. Mice Inbred Strains 

C57BL/6 mice and B1R^−/−^ transgenic mice were maintained in the animal facilities of the Biotério de Experimentação em Imunobiologia (BEI/UFRJ). Mice were maintained under specific pathogen-free (SPF) conditions with ad libitum water and free access to standard chow. All sets of experiments were conducted with sex- and age-matched mice, weighing 20–25 g and housed in micro-isolators (Alesco CO., São Paulo, Brazil) at 22 ± 2 °C with a 12 h light dark cycle. Animal care and experimental procedures were carried out in accordance with current guidelines for experiments in conscious animals and approved by the ethical committees of UFRJ (code number: 048/2019).

### 2.2. Source of T. cruzi and Experimental Infection

Tissue culture trypomastigotes (TCTs) Dm28c were harvested from the supernatants of cultures of monkey kidney fibroblast cell line LLCMK2 cultures. The cultures were maintained in Dulbecco’s Modified Eagle Medium (DMEM), 2% heat inactivated fetal calf serum (FCS). The freshly released TCTs were washed twice with excess Hank’s Balanced Saline Solution (HBSS) before being injected. Male C57BL/6 mice were injected intraperitoneally with Dm28c TCTs (10^3^ or 10^6^). At 15, 60, or 90 days after the onset of infection, the mice were euthanized, and different organs were removed and processed for histopathological analyses. A second model of infection involved intraperitoneal inoculation of 100 blood trypomastigotes of the Colombian *T. cruzi* strain into C57BL/6 mice. Here we used females because this gender is more resistant to acute *T. cruzi* infection [45,46]. Blood levels of Col trypomastigotes were quantified in samples of tail vein blood and the survival rate was registered weekly. At 60 or 160 days after infection, mice were euthanized and different organs were removed and processed for histopathological analyzes.

### 2.3. Quantitative Determination of Kinin B1 Receptor Levels and T. cruzi DNA in Cardiac Tissues by Real Time Polymerase Chain Reaction (RT-PCR)

C57BL/6 mice were injected intraperitoneally with Dm28c TCTs (10^6^). After 15 days of infection, total RNA of heart tissues was isolated using the total RNA isolation system Kit (Promega, Madison, WI, USA) following the manufacturer’s protocol. Quantitative gene expression of B1R was determined using the SYBR-green fluorescence quantification system. RNA was reverse transcribed from 3 μg of total RNA in a final volume of 20 μl using Superscript II transcriptase (Invitrogen, Waltham, MA, USA, No 18064014) according to the manufacturer’s protocol. Real PCR was performed using 100 ng cDNA, and 0.1 μM of each primer (sense and antisense) in a final volume of 20 μL. The reaction was incubated for 50 °C for 2 min, 95 °C for 10 min, and 50 cycles of 95 °C for 15s and 60 °C for 1 min. The fluorescence was detected at the end of each cycle at the ABI 7000 (Applied Biosystems). The primers sequence used for B1R were: 5′ CCATAGCAGAAATCTACCTGGCTAAC 3′ sense; 5′ GCCAGTTGAAACGGTTCC 3′ antisense, generating a PCR band of 102 bp. The primer sequences used for GAPDH were: 5′ CTCCCACTCTTCCACCTTCG 3′ sense; 5′ GCCTCTCTTGCTCAGTGTCC 3′ antisense, generating a PCR band of 189 bp.

The quantitative RT-PCR was used to determine the parasite DNA content in cardiac tissue of mice. The DNA was isolated using the DNeasy blood and tissue Kit (QIAGEN, Venlo, The Netherland, No 69504) or PureLink genomic DNA miniKit (Invitrogen, Waltham, MA, USA, No K182002) according to the manufacturer’s instructions. The PCR for parasite quantification was performed using 20 ng or 100 ng of total DNA as previously described [47,48,49]. The SYBR-green fluorescence quantification system was used and the standard PCR conditions were 95 °C (10 min), and then 45 cycles of 95 °C (30 s) and 60 °C (1 min), followed by the standard denaturation curve. Each reaction contained 0.4 μM of each primer of *T. cruzi* specific primers: GCTCTTGCCCACACGGGTGC (forward) and CCAAGCAGCGGATAGTTCAGG (reverse); and 0.1 μM of each primer for genomic B2m, CTGAGCTCTGTTTTCGTCTG (forward) and TATCAGTCTCAGTGGGGGTG (reverse). Measurements of *T. cruzi* DNA content was normalized using the ratio of Ct values for *T. cruzi*- and mouse-specific PCR and converted to estimated numbers of parasite equivalents by reference to a standard curve with a range of 2.5 × 10^5^ − 2.5 parasites equivalents. The standard curve was established from serial dilution (10×) of a *T. cruzi* DNA. The reactions were performed in a StepOne (Applied Biosystems, Warrington, UK) sequence detection system with Power SYBR^TM^ Green PCR Master Mix (Applied Biosystems, Warrington, UK, No 4309155) according to the manufacturer’s recommendations. 

### 2.4. Flow Cytometry Analysis

C57BL/6 and B1R^−/−^ mice were infected with Dm28c TCTs (10^6^ parasites; i.p.). At 15 dpi, the mice were euthanized and the cardiac tissue was processed. The hearts were perfused with 10 mL of PBS after a cut to the superior vena cava, and then cut into small pieces and digested in solution containing DNAse (0.1 mg/mL; Sigma-Aldrich, St. Louis, MO, USA) and collagenase (1 mg/mL; Sigma-Aldrich) for 1 h at 37 °C. For monocytes/neutrophils analysis, Fc receptors were blocked with anti-CD16/CD32 (BD Biosciences, New York, NY, USA), cells were stained with anti-CD11b-PECy7 (Biolegend, San Diego, CA, USA), anti-Gr1-FITC BD Biosciences, anti-Ly6C-APC (Biolegend), and anti-F4/80-PE antibodies (Biolegend). Cells were acquired with a FACSCalibur (BD Biosciences, New York, NY, USA) and the data were processed with Summit software (DAKO, Colorado, Inc., Fort Collins, CO, USA). Flow cytometry analyses were performed at Plataforma de Imuno-Análise (PIA, IBCCF, UFRJ, Brazil), Centro Multiusuário Darcy Fontoura de Almeida (CMDFA, IBCCF, UFRJ, Brazil), IBCCF/UFRJ.

### 2.5. Histopathological Analysis of Heart Tissues

Dm28c-infected WT and B1R^−/−^ mice and Colombian-infected C57BL/6 mice were euthanized after 90 or 160 dpi, respectively, and the total hearts were fixed in formalin 5% for 24 h at room temperature. Hearts (cutting in half) were transferred to the tissue cassette and dehydrated through serial EtOH incubation for 30 min. They were then clarified in xylene (three times) for 30 min each. After one hour in Erv-plast paraffin (Easypath-Brazil) at 58 °C, hearts were embedded in the same paraffin. Organ sections (4 μm) were removed from paraffin and incubated in serial EtOH for hematoxylin/eosin (HE) (Sigma-Aldrich, MO, USA) or picrosirius staining technique in order to evaluate infiltrating cells and collagen fibers percentage, respectively. Following HE staining, the intracardiac infiltrating cells were detected using a LEICA scan microscope with 20× objective (±200 cells in 50 fields). In Picrosirius Red staining, images of the total heart were analyzed with 10× objective. Seven sections of each half of the heart were examined and represented as percentage of total collagen area. The morphometric analyses of collagen deposition were performed using Image Pro-Plus 6.0 (Image Processing and Analysis Software). Hearts from non-infected mice used as controls did not show pathologic changes. 

### 2.6. Determination of Parasite Nests by Immunohistochemistry

At 60 days post infection, Colombian-infected mice were euthanized, the hearts were removed, and the tissue was embedded in tissue-freezing medium (Tissue-Tek, Miles Laboratories, Elkhart, IN, USA) and stored in liquid nitrogen for analysis by immunohistochemistry. Serial cryostat sections, 3 mm thick, were fixed in cold acetone and subjected to indirect immunoperoxidase staining. The presence of *T. cruzi* nests in cardiac tissue was quantified by using a polyclonal antibody against *T. cruzi* antigens (produced in LBI/IOC-Fiocruz, Brazil), using a digital morphometric apparatus. The images were analyzed using the AnaliSYS Program and the areas containing parasite molecules were identified as amastigote nests. For each heart sample, three separate tissue sections and 50 fields per section were analyzed. The number of amastigote nests was determined in 100 microscopic fields (magnification 400×) per tissue section.

### 2.7. Electrocardiographic Measurements

Analysis at 60, 120, and 150 dpi: Mice were tranquilized with diazepam (10 mg/kg) and electrodes were placed subcutaneously (DII). The traces were recorded for 2 min using a digital Power Lab2/20system connected to a bio-amplifier (PanLab Instruments, Spain). The filters were set at 0.1 and 100 Hz and the traces were analyzed using Scope software for Windows V3.6.10 (PanLab Instruments, Spain). The ECG parameters were analyzed as previously described [17].

Analysis at 120 and 160 dpi: Two stainless steel electrodes were placed subcutaneously in lead I configuration. Electrodes were connected by flexible cables to a differential AC amplifier (model 1700, A-M Systems, USA), with signals low-pass filtered at 50 Hz and digitized at 2–10 kHz sample rate by a 16-bit A/D converter (Digidata 1322-A, Axon Instruments, Union City, CA, USA) using Axoscope 9.0 software (Axon Instruments, Union City, CA, USA). Data were stored on a PC for offline processing, using Axoscope. All recordings were conducted in a quiet environment during morning hours (07:00–11:00 h).

### 2.8. Creatine-Kinase MB Detection

The activity of the creatine kinase (CK-MB) in the serum was measured using a commercial kit (LabTest, Minas Gerais, Brazil, No 118-2-30). The assay was adapted for reading in a microplate and performed according to the manufacturer’s recommendations. The optical density at 340 nm was recorded every 5 min for 15 min and the results are expressed by fold change.

### 2.9. Cell Cultures and Invasion Assays with T. cruzi

Human primary umbilical vein endothelial cells (HUVECs) were obtained by treatment of umbilical veins with a 0.1% (wt/vol) collagenase IV solution. Primary HUVECs were seeded in 25 cm^2^ flasks coated with 2% porcine skin gelatin and grown in M199 medium supplemented with 2 mM glutamine, 2.5 µg/mL amphotericin B, 100 µg/mL penicillin, 100 µg/mL gentamycin, 0.13% sodium bicarbonate, and 20% FCS. Cells were maintained at 37 °C in a humidified 5% CO_2_ atmosphere until confluence. For invasion assays, monolayers of HUVECs were prepared into 24-well plates with gelatin-coated glass coverslips (13 mm round) at a density of 5 × 10^4^ cells in culture medium supplemented with 10% inactivated FCS. Before being incubated with parasites, the monolayers were washed with HBSS and kept in the appropriate serum-free culture medium containing 1 mg/mL of human serum albumin (HSA) (Baxter Pharmaceutical, Deerfield, IL, USA) and 25 µM of lisinopril (ACE inhibitor; Sigma) diluted in saline. Parasite–host cell interaction took place in a final volume of 300 µL per well. Before addition of Dm28c or Colombian TCTs (ratio of 20:1 parasite/host cell), the culture medium was supplemented, or not, with the B2R antagonist (HOE-140; 0.1 µM; Sigma) or the B1R antagonist [Leu^8^]des-Arg^9^-BK at 1 µM. Host–parasite interactions proceeded for 2 or 24 h at 37 °C in a humidified incubator in a 5% CO_2_ atmosphere. The involvement of the major cysteine protease of *T. cruzi* (cruzipain) was tested by pretreating the washed TCTs with K11777 at 10 µM for 20 min versus treatment with DMSO (0.025%) in PBS. After washing, the suspension of TCTs was added to the HUVEC monolayers. At the indicated time point, the cultures were washed three times with HBSS, once with PBS, then fixed with Bouin and stained with Giemsa. The efficiency of infection at 2 h or 24 h was measured as number of intracellular parasites in a total of 100 cells per coverslip. 

### 2.10. ELISA for IFN-γ Detection

The concentrations of IFN-γ in the splenocytes culture were evaluated by enzyme-linked immunosorbent assay (ELISA) DuoSet kits (R & D Systems, Minneapolis, MN, USA) according to manufacturer’s instructions. 

### 2.11. Intracellular Staining for IFN-γ and Granzyme B Detection

C57BL/6 and B1R^−/−^ mice were infected with Dm28c TCTs (10^6^; i.p.). After 15 days, the animals were euthanized, and the spleen and heart were removed. The hearts were perfused with 10 mL of PBS after an incision to the superior vena cava, cut into small pieces and digested in DNAse (0.1 mg/mL; Sigma) and collagenase (1 mg/mL; Sigma) solution for 1 h at 37 °C. Splenocytes and cardiac cells were incubated with monensin (500 nM) for 8–10 hat 37 °C in the presence of a soluble extract of *T. cruzi* epimastigotes (50 μg/mL) plus H-2Kb-restrictedTskb20 peptide (ANYKFTLV; 5 μM). The cells were collected, washed, resuspended in PBS containing 2% fetal calf serum and sodium azide 0.1% (FACS buffer) and then labeled for flow cytometry analysis. After Fc receptors were blocked with anti-CD16/CD32 (BD Biosciences), cells were stained with anti-CD8-FITC (Biolegend) and anti-CD4- PECy7 (Biolegend), fixed for 1 h in paraformaldehyde 1% at RT, and permeabilized for 20 min with FACS buffer containing 0.2% saponin (Sigma-Aldrich). Intracellular staining was performed with anti-IFN-γ-PE (BD Biosciences) and anti-Granzyme B-A647 (Biolegend) antibodies. At least 20,000 gated CD4^+^ cells or CD8^+^ cells were acquired.

### 2.12. In Vivo Cytotoxicity Assay

Splenocytes from naive C57BL/6 mice were divided into two populations and incubated with H-2Kb-restricted TsKb20 peptide *T. cruzi*-derived for 40 min at 37 °C (target cells) or unloaded (control), as previously described [48]. These populations were then labeled with two concentrations of carboxyfluorescein diacetate succinimidyl diester (CFSE), respectively: 5 μM (CFSE^high^) or 0.5 μM (CFSE^low^). CFSE^high^ target cells were washed and mixed with equal numbers of CFSE^low^ cells and injected iv. at concentration of 10^7^ cells into *T. cruzi*-infected and non-infected control mice. After 18 h, spleens were collected and the samples were analyzed by flow cytometer. For analysis, 30,000 events were acquired with a FACSCalibur (BD Biosciences) and the data were processed using Summit v.4.3 Build 2445 software (Dako, Glostrup, Denmark). The percentage of specific lysis was determined using the formula: 1 − [%CFSE^high^ infected/%CFSE^low^ infected]/[%CFSE^high^ noninfected/%CFSE^low^ noninfected] × 100%.

### 2.13. Reagents 

Source of B1R antagonists: [Leu^8^]des-Arg^9^-BK (Sigma, St. Louis, MI, USA); R-954 was supplied by one of the co-authors (PS) [50].

### 2.14. Statistical Analyses 

The student t test or analysis of variance (ANOVA) was used to determine statistical significance (Prism, GraphPad) followed by Tukey’s multiple comparison tests. Statistical significance was set at *p* < 0.05. *p* values are shown in the graphs. 

## 3. Results

### 3.1. Global Ablation of the Mouse b1r Gene Reduces Heart Parasitism and Inhibits Intracardiac Infiltration of Proinflammatory Leukocytes in the Acute Stage of CD 

Since B2R knockout mice were susceptible to acute *T. cruzi* infection (Dm28 strain) [25], in the current study we aimed to determine whether the outcome of infection was similar or different in mice in which the *b1r* gene was globally ablated. To this end, WT (C57BL/6) and B1R^−/−^ mice were challenged with a high dose inoculum of Dm28c TCTs (10^6^ parasites). As shown in the previous study [25], we measured the levels of mRNA of B1R at 15 dpi, a time point in which heart tissues from WT C57BL/6 mice are already parasitized. As predicted, the transcriptional expression of the *b1r* gene in WT animals was vigorously upregulated as compared to the baseline mRNA levels observed in non-infected controls (Figure 1A; 280% upregulation; *p* = 0.0033). 

We then compared the *T. cruzi* DNA levels in the heart tissues of both mice strains (15 dpi) and found that heart parasitism was markedly decreased in B1R^−/−^ mice as compared to WT littermates (Figure 1B; 400% reduction; *p* = 0.0006). Since heart parasitism was decreased in acutely infected B1R^−/−^ mice, we then asked whether heart-infiltration (15 dpi) by circulating neutrophils and inflammatory monocytes was influenced by presence/absence of B1R. Starting the analysis with the total CD11b^+^ population (Figure 1C), we found that heart-infiltrating neutrophils (CD11b^+^Gr1^hi^Ly6C^int^) and inflammatory monocytes (CD11b^+^F4/80^−^Gr1^low^Ly6C^hi^) were significantly reduced in B1R^−/−^ infected mice compared to the WT infected mice (Figure 1D–F; 35.3% and 37.7% reduction; *p* = 0.0199 and *p* = 0.0231, respectively). 

Since the B2R/IL-12 pathway was previously implicated in the intralymphoid (splenic) development of immunoprotective (IFN-γ-expressing) effector T cells [25], we were concerned that the cardioprotective phenotype of B1R^−/−^ mice could reflect enhanced intracardiac clearance of the parasites due to compensatory upregulation of B2R. However, arguing against this potential caveat, flow cytometer analysis did not detect significant differences in the intracardiac frequencies of granzyme B (GzB)- and IFN-γ-producing CD8^+^ and CD4^+^ T cells of WT and transgenic mice (Appendix A). Along similar lines, we did not find differences in GzB and IFN-γ expressing T cells isolated from the spleen of acutely infected WT and B1R^−/−^ mice (Appendix A). Using ELISA, we then compared the IFN-γ secretion by splenocytes from WT and B1R knockout infected mice exposed to soluble epimastigote extracts or the immunoprotective H-2Kb-restricted TsKb20 peptide. In both cases the response was not enhanced in infected B1R^−/−^ mice (Appendix A). Extending the breadth of these studies to the in vivo settings, we next measured CD8 T cell-mediated cytotoxicity in vivo by transferring into infected WT or B1R^−/−^ mice a fixed number of target cells loaded with H-2Kb-restricted TsKb20, an immunodominant peptide antigen from *T. cruzi* [51]. After isolating splenic cells from recipient WT and B1R^−/−^ mice (18 h after cell transfer), we found that target cytotoxicity was not enhanced in B1R-deficient mice (Appendix A). In summary, there is no evidence that compensatory upregulation of type-1 effector immunity may account for the cardioprotective phenotype of B1R^−/−^ mice. 

Next, we checked whether the cardioprotective phenotype of B1R-deficient mice was maintained as the infection progressed towards the chronic phase. In a set of experiments performed at 60 dpi, we confirmed that levels of *T. cruzi* DNA were vigorously reduced in B1R^−/−^ mice (Figure 1G; 85.7% reduction; *p* = 0.03). Of further interest, the cardioprotective phenotype of B1R^−/−^ mice was further substantiated by the evidence that creatine kinase-MB (CK-MB), a biomarker of cardiac muscle injury, was exclusively detected in the serum of WT infected mice (Figure 1H; *p* = 0.01). 

### 3.2. Chronic Myocarditis and Heart Fibrosis Are Attenuated in B1R^−/−^ Infected Mice

In order to investigate the role of the KKS in the pathogenesis of CCC, WT and B1R^−/−^ mice were infected with a sub-lethal dose of Dm28c TCTs (10^3^ parasites). Ninety days later, the hearts of chronically infected mice were isolated and subjected to qPCR (*T. cruzi* DNA) and histopathological analysis. Since the intracardiac levels of *T. cruzi* DNA were below detection in both mice strains, we surmise that the intracardiac load (low-grade) was efficiently cleared by adaptive immunity, irrespective of presence or absence of B1R. The images and the accompanying graph revealed that WT-infected mice displayed a prominent chronic myocarditis (90 dpi) in contrast to the scant inflammatory infiltrates detected in heart tissues from B1R^−/−^ mice (Figure 2A, 86.2% reduction; *p* = 0004). We next examined the extent of heart fibrosis in the chronically infected mice by Picrosirius red staining. Consistent with the milder chronic myocarditis observed in the B1R^−/−^ infected mice, collagen deposition in the transgenic heart was decreased as compared to WT infected mice (Figure 2B, 64% reduction; *p* = 0.01).

In order to evaluate the therapeutic potential of R-954, a well-characterized antagonist of B1R [50], it was preferable to use a classical mouse model of chagasic cardiomyopathy: C57BL/6 mice chronically infected by the myotropic Colombian (Col) strain [52]. Before starting the in vivo studies with R-954, we checked whether the Colombian strain (Col TCTs) shares with Dm28c TCTs the capacity to invade endothelial cells via the activation of B2R and/or B1R [53,54]. To this end, monolayers of primary cultures of human (resting) umbilical vein endothelial cells (HUVECs) were incubated with Col TCTs or Dm28c TCTs for 2 h in serum-free medium containing the ACE inhibitor lisinopril. Unlike the previously described phenotype of Dm28c TCTs [53], which readily invaded resting HUVECs via activation of the B2R pathway (Appendix A), Col TCTs were not able to infect these monolayers within 2 h (Appendix A). However, given the precedent that HUVECs activated by LPS were infected by Dm28c TCTs via the B1R pathway [54], the incubation time of HUVECs and Col TCTs was extended to 24 h. Strikingly, we found that Col TCTs were vigorously internalized by activated HUVECs (Appendix A). Importantly, the invasion was blocked in culture medium supplemented with the B1R antagonist [Leu^8^] des-Arg^9^-BK (Appendix A). Collectively, these in vitro studies suggest that Col TCTs might exploit the upregulated surface expression of B1R to increment its infectivity, reminiscent of the infective phenotype of Dm28c TCTs [54]. Of further interest, we found that Col TCTs pretreated with K11777, an irreversible inhibitor of cruzipain [55], reduced the B1R-dependent internalization of the pathogen (Appendix A). 

After being assured that Col TCTs infect HUVECs (24 h) via the cruzipain/B1R pathway, we then challenged C57BL/6 mice (i.p.) with a low inoculum of Col TCTs. Starting at 15 dpi, R-954 was administered daily for 45 days (1.6 mg/kg, s.c—Figure 3A). Interestingly, R-954 (60 dpi) did not reduce the blood parasitemia (Figure 3B), a parameter that reflects the extent of parasite load in several tissues, mostly the spleen, at early stages of infection. However, measurements of the number of intracardiac nests of parasites revealed that R-954 consistently reduced heart parasitism at the end of the drug treatment (60 dpi.) (Figure 3C; 43% reduction; *p* = 0.004). 

Next, we interrogated whether R-954 treatment (15–60 dpi) had an impact on cardiac function of the acutely infected mice. ECG records in sedated mice made at 60 dpi showed a reduction in the heart rate of infected mice (≈437 bmp) compared to the non-infected (NI) group (≈530 bmp) (Figure 3D; 17.5% reduction; *p* = 0.0129). Although the treatment with R-954 was initiated 45 days earlier, the B1R blocker did not reverse the heart rate alterations induced by the acute infection (Figure 3D; ≈443 bmp; 16.4% reduction in comparison to NI mice; *p* = 0.02). Likewise, the PR interval, which was not significantly increased at 60 dpi (NI ≈ 39 ms; *T. cruzi* + vehicle ≈ 44 ms) (Figure 3D), did not change upon R-954 treatment (*T. cruzi* + R-954 ≈ 42 ms). The QRS duration, which reflects the duration of the depolarization of ventricles, did not differ between *T. cruzi-*infected mice (60 dpi) and controls (NI ≈ 11 ms; *T. cruzi* + vehicle ≈ 11 ms). However, R-954 slightly reduced the duration of QRS complex registered in the infected (60 dpi) mice (*T. cruzi* + vehicle ≈ 11 ms; *T. cruzi* + R-954 ≈ 10 ms; *p* = 0.043). The duration of the QT interval, corrected by Bazett’s formula (QTc), increased in infected mice (60 dpi) as compared to the NI group (NI ≈ 78 ms; *T. cruzi* + vehicle ≈ 106 ms; *p* = 0.0003) (Figure 3D), but this alteration was not reversed by R-954 (*T. cruzi* + R-954 ≈ 98 ms; *p* = 0.0062 in comparison to NI group). A second-degree atrioventricular block (AVB2) was observed in 67% of *T. cruzi*-infected mice (60 dpi.), while the NI animals did not present AVB2, as expected. The incidence of AVB2 in infected animals (60 dpi) was reduced (62%) by treatment with R-954 (Figure 3D). Furthermore, the incidence of arrhythmias in *T. cruzi*-infected mice (60 dpi) was approximately 70% whereas R-954 treatment reduced its frequency to approximately 57% (Figure 3D). In line with infection-driven ECG alterations, we detected serum CK-MB activity at 60 dpi (1.8-fold change; *p* = 0.04). Recapitulating the cardioprotective phenotype of B1R^−/−^ mice infected by the Dm28c strain (Figure 1H), the rise of serum CK-MB levels was reversed by the B1R antagonist (Figure 3E; 34% reduction in comparison to untreated mice; *p* = 0.02). In conclusion, R-954 treatment during the acute stage of infection (45 days) reduced the extent of heart parasitism and attenuated cardiac tissue injury. However, the B1R blocker did not reverse heart rates and had modest cardioprotective effects on the incidence of cardiac arrythmias. 

### 3.3. Therapeutical Effects of R-954 during Chronic Chagas Disease 

Since the B1R antagonist reduced the extent of heart parasitism during the acute stage of infection, we then shifted the time window of R-954 treatment to the chronic stage of infection (120–160 dpi; Figure 4A), hence, covering the period in which heart injury consistently causes electrical disturbances [52] (Figure 4B). In this procedure, C57BL/6 mice infected with the Colombian strain were either treated daily for 40 days with vehicle (saline) or R-954. The first evidence that R-954 treatment had therapeutic potential was that the lethality index was reduced compared to those of untreated infected mice (Figure 4C; 72% and 95% of survival, vehicle versus R-954-treated mice, respectively; *p* = 0.033). 

We then compared EGG parameters in the chronic stage of infection, using two different experimental conditions: sedated (Appendix A) and awake mice (Figure 4). Measurement in the absence of diazepam-sedation showed that heart rate at 160 dpi was reduced in relation to non-infected (NI) mice (≈706 bpm versus ≈792 bpm, respectively; *p* = 0.0192) and this alteration was not reversed by the B1R blocker (*T. cruzi* + R-954 ≈ 696 bpm) (Figure 4D). Similar to results obtained at 120 dpi (Figure 4B), the PR interval of non-sedated mice was reduced at 160 dpi (NI ≈ 32 ms; *T. cruzi* + vehicle ≈ 24 ms; *p* = 0.048) but this response was not reversed by R-954 treatment (Figure 4D; *T. cruzi* + R-954 ≈ 25 ms). Interestingly, however, measurements of the QRS duration (which did not change at 120 dpi), was prolonged at 160 dpi compared to the NI group (NI ≈ 9 ms; *T. cruzi* + vehicle ≈ 29 ms; *p* = 0.03). However, this ECG abnormality was reversed by R-954 treatment (*T. cruzi* + R-954 ≈ 11 ms; *p* = 0.041 in comparison to the untreated group) (Figure 4D). Consistent with ECG registers made at 120 dpi (Figure 4B), the QTc values at 160 dpi were also prolonged (NI ≈ 13 ms; *T. cruzi* + vehicle ≈ 16 ms; *p* = 0.04), and this alteration observed in infected mice was reversed by R-954 treatment (*T. cruzi* + R-954 ≈ 12 ms; *p* = 0.008 in comparison to untreated group) (Figure 4D). We then measured second-degree atrioventricular blocks (AVB2), which only affected 4% of the infected non-sedated mice at 120 dpi (Figure 4B). The AVB2 response increased to 10% at 160 dpi. Following R-954 treatment, however, AVB2 incidence at 160 dpi was reduced to 8% (Figure 4D). The incidence of arrhythmias at 120 and 160 dpi was 41% and 34%, respectively, but the B1R blocker only had a minor impact on these parameters (29%) (Figure 4D). Chronic infection with Col *T. cruzi* induced far more profound ECG alterations in diazepam-sedated mice as compared to non-sedated animals, including a discrepant rise in PR interval (Appendix A). Importantly, however, the infection-induced alteration of PR intervals was reverted by R-954. 

Next, we performed histopathological studies (160 dpi.) to evaluate whether R-954 treatment modulated chronic myocarditis and heart fibrosis (Figure 4E–G) and found that 40 days treatment with the B1R blocker efficiently reduced the extent of inflammatory infiltration in the heart tissues (Figure 4F; 36.4% reduction in comparison to untreated group; *p* = 0.0001). Using Picrosirius red staining, we verified that R-954 reduced collagen deposition in the majority (80%) of mice (Figure 4G), but statistical significance was not achieved. Although the treatment with R-954 reversed the rise of serum CK-MB levels during the transition from acute to chronic infection (60 dpi; Figure 3E), treatment with the B1R blocker (120–160 dpi) did not reverse the cardiac injury at the endpoint of the study (Figure 4H).

## 4. Discussion

Having evolved for millions of years in the sylvatic environment, *T. cruzi* has developed overlapping strategies to persistently infect immunocompetent mammalian hosts [12,56]. Although advances in immunology and molecular parasitology improved our knowledge about the pathogenesis of CD [10,16,17,18], technical obstacles have precluded analysis of microvascular dysfunctions during the transition from acute to chronic stage of CD. In spite of these limitations, intravital microscopy studies in hamster cheek pouch tissues provided evidence that inflammatory neovascularization, a dynamic process involving the activation of proteolytic cascades, continuously translates into mutual benefits to the host/parasite relationship [9,13,14,31]. Using a combination of animal models of acute CD, we recently demonstrated that BK-induced inflammatory cascades fuel heart parasitism via crosstalk between B2R and endothelin receptors [14]. As stated in the introduction, there is a major difference in the mechanisms whereby B2R and B1R regulate vascular homeostasis. While BK induces de-sensitization of B2R in the endothelial lining, the interaction of DABK with the B1R stabilizes the surface of B1R in activated endothelial cells [57]. Consistent with these mechanistic principles, mice inoculated with Dm28c TCTs developed an initial inflammatory edema controlled via the BK/B2 pathway, which is subsequently prolonged via the activation of the pro-inflammatory DABK/B1R pathway [54]. 

Using two different models of systemic infection with *T. cruzi*, in the current study, we provide evidence that the activation of the pro-inflammatory DABK/B1R pathway is detrimental to heart pathology in CD. Briefly recapitulating our main findings, we found that B1R was transcriptionally upregulated in the WT heart at 15 dpi, a time point in which the intracardiac load of Dm28c *T. cruzi* (15 dpi) was drastically increased as compared to B1R^−/−^ mice. Analyses by flow cytometry revealed that frequencies of pro-inflammatory neutrophils and monocytes were increased in the WT heart. As the infection progressed, *T. cruzi* DNA levels (60 dpi) remained lower in cardiac tissues of B1R^−/−^ mice. Moreover, CK-MB activity (60 dpi) was exclusively detected in the serum of acutely infected WT mice, implying that cardiac injury was attenuated in the absence of B1R. Finally, histopathological analysis revealed that chronic myocarditis and heart fibrosis (90 dpi) were markedly attenuated in B1R^−/−^ mice, confirming that B1R plays a detrimental role in heart pathology. 

At first sight, the cardioprotective phenotype of B1R^−/−^ mice described in our work is reminiscent of findings reported in a mouse model of diabetic cardiomyopathy induced by streptozotocin [44]. Admittedly, however, our findings cannot be easily reconciled with evidence that activation of the KKS is cardioprotective in other mouse models of heart injuries, whether involving the constitutive B2R or the inducible B1R pathway [41,42,43,58]. Although the immunoprotective role of B2R in acute CD is unequivocal [25], the assessment of B1R function in mouse models of sterile heart injuries is limited because they lack the T cell-dependent component of chronic heart pathology that underlies CCC. 

There are several limitations in our study, mostly related to the fact that we have not investigated the impact of *T. cruzi* infection in transgenic mice with conditional deficiency of B1R in specific cell types. Since adipocytes constitutively express B1R, the possibility that *T. cruzi* may exploit this exceptional feature of adipose tissues comes to mind because adipocytes were previously identified as a privileged niche for intracellular outgrowth of *T. cruzi* [59]. In the absence of B1R, adipose tissues are not efficiently expanded, and hepatic lipid production is compromised, at least so in models of diet-induced obesity [60]. Studies in transgenic mice with selective deficiency of B1R in adipocytes may clarify whether *T. cruzi* invasions of heart tissues are preceded by intracellular cycles of infection in the pericardial adipose tissue or coronary arteries. 

Strictly analyzed from the perspective of experimental design, the first part of our study suffered from the fact that global ablation of *b2r* or *b1r* genes tends to overexpress B1R or B2R in a compensatory manner [61]. Arguably, we cannot rule out the possibility that B2R overexpression in the endothelium had a partial contribution to the cardioprotective phenotype of *T. cruzi*-infected B1R^−/−^ mice [62]. However, the differences in the extent of heart parasitism in WT and B1R^−/−^ mice should have been narrowed if *T. cruzi* trypomastigotes had profited from compensatory overexpression of B2R in the inflamed heart tissues of the transgenic mice [14]. A second point of concern was that B2R overexpression in our transgenic line could in principle enhance intracardiac clearance of the parasites, based on the premise that BK, acting as a driver of IL-12 production by B2R-expressing (splenic) dendritic cells, promotes the development of immunoprotective (IFN-γ -producing) effector T cells [24,25]. In a series of experiments, we ruled out this potential caveat by showing that splenic effector T cells from acutely infected WT and B1R^−/−^ mice produced similar levels of IFN-γ, whether stimulated by crude *T. cruzi* antigens or MHC-class I-restricted *T. cruzi* peptides. Moreover, we did not detect differences in the frequencies of INF-γ-producing T cells nor GzB-expressing CD8^+^ T cells infiltrating the heart of WT and B1R^−/−^ mice. Future studies using mice with genetic deficiency of B1R limited to the endothelium may clarify whether the activation of the KKS/B1R pathway may enhance TNF/TNFR1-dependent heart infiltration by CCR5^+^CD8^+^ effector T cells [63]. Reciprocally, there is an urge to evaluate whether the intracardiac activation of pathogenic subsets of effector CD8 T cells [17] might compromise ACE2-dependent regulation of the pro-inflammatory B1R pathway [38].

Since the egress of trypomastigotes from dying host cells is an asynchronous process [8,9], the transcriptional upregulation of B1R in heart tissues can be triggered by a myriad of inflammatory cues, including IL-1β generated upon inflammatory death of heart cells and/or pro-inflammatory lipid anchors shed by extracellular trypomastigotes [30,55]. Interestingly, resting HUVECs were not readily infected by Col TCTs, unlike Dm28c TCTs, which rapidly (2 h) invaded endothelial cells via activation of the cruzipain/kinin/B2R pathway [53]. However, Col TCTs were vigorously internalized via the B1R pathway when the incubation was prolonged for several hours, thus, recapitulating the phenotype of Dm28c TCTs in monolayers of HUVECs activated by lipopolysaccharide (LPS) [54]. Although speculative, it is conceivable that some strains of *T. cruzi* might have differential ability to upregulate B1R via t-GPI-dependent activation of the TLR2 pathway [30,55]. 

Involving C57BL/6 mice infected systemically by the Colombian strain, our pharmacological studies revealed that R-954 treatment during the acute phase (15–60 dpi) consistently reduced the number of pseudocysts in heart tissues. Noteworthy, we verified that serum levels of CK-MB were exclusively detected in acutely infected (untreated) mice, implying that the B1R antagonist mitigated cardiac injury caused by the intracellular parasite. Since inflamed heart tissues contain several types of cells that might be susceptible to *T. cruzi* infection (cardiomyocytes, vascular endothelial cells, cardiac fibroblasts, smooth muscle cells, macrophages, and stem cells recruited to the injured heart tissues), it remains to be determined whether R-954 inhibited DABK-mediated internalization of trypomastigotes by particular subsets of B1-expressing heart cells or inhibited the invasion of a wide range of targets upregulating B1R. Alternatively, the B1R antagonist might have reduced *T. cruzi* infectivity by limiting plasma leakage and the proteolytic formation of infection-inducing peptides, such as BK and DABK [54], at the downstream end of the inflammatory cascade. 

In a seemingly contradictory finding, we verified that R-954 treatment failed to reduce the blood parasitemia during the acute phase of infection. Since macrophages are the main targets of *T. cruzi* infection in the spleen during the acute phase, the levels of blood parasitemia are dominantly influenced by the extent of parasite burden in splenic macrophages. Unlike the microcirculation of the heart and gastro-intestinal tissues, which are lined by non-fenestrated endothelium, plasma fluids diffuse freely through the sinusoids of the spleen and liver. As acute infection progresses, the phagocytic uptake of trypomastigotes by splenic macrophages may be vigorously stimulated as a result of free diffusion of plasma-borne opsonins, such as IgG antibodies and complement system factors, through the fenestrated endothelium of splenic sinusoids. Under the circumstances, it is unlikely that pharmacological targeting of B1R might interfere with the phagocytic uptake of *T. cruzi* by splenic macrophages and this may explain why R-954 failed to reduce blood parasitemia during acute infection. 

Lasting several years, the low-grade intracardiac infection may gradually cause inflammatory lesions that translate into upregulated expression of B1R in cardiac micro vessels. Sporadically, a productive cycle of parasite outgrowth in B1R-expressing endothelial cells may cause microvascular injury, similar to the heart microangiopathy described in *T. cruzi*-infected dogs [22]. Potentially aggravated by platelet activation and microthrombi formation within damaged capillaries, the endothelial injury must be promptly repaired via stimulation of angiogenesis. Interestingly, recent studies of the dynamics of inflammatory neovascularization in the parasitized tissues (hamster cheek pouch) revealed that vascular remodeling is fueled by chymase, a proangiogenic/fibrotic serine protease released from mast cell secretory granules [9]. Strategically localized in the subendothelial matrix, mast cells may further promote vascular and tissue remodeling by activating the proangiogenic B1R pathway [13,44]. Studies in animal models of CCC may clarify whether the dysfunctional regulation of the proangiogenic KKS/DABK/B1R pathway [21,58] may cause excessive nitric-oxide-dependent vasodilation, perhaps reproducing the microcirculatory derangement described in the myocardium of CCC patients [21]. 

As stated earlier, it is possible that a prolonged destabilization of the endothelial barrier forged via the activation of the KKS/DABK/B1R pathway might translate into reciprocal benefits to the host/parasite relationship [13,14]. Nisimura et al. have recently reported that angiogenesis parallels cardiac tissue remodeling in mice acutely infected by *T. cruzi* [64]. Interestingly in this context, we recently showed evidence that microvascular permeability is subtly increased at the early stages of infection in the hamster cheek pouch, preceding the vascular remodeling process that underlies angiogenesis [9]. Based on these findings, we postulated that parasite-induced opening of the endothelial “floodgates” may accelerate the delivery of blood-borne nutrients and survival factors to the foci of infection, hence, meeting the metabolic demands of parasitized host cells [9]. Reciprocally benefiting the parasitized/injured host, a sustained extravasation of plasma via B1R may provide extravascular spaces with the coagulation factors that drive the formation of the provisional fibrin matrix that supports the migration of endothelial tip cells during angiogenesis [65]. 

Although heart rates were not reversed by R-954 during the acute infection (60 dpi), the clinical benefits of B1R blockade were more persuasive when we shifted the drug treatment to the chronic stage of Col infection (120–160 dpi). During this advanced period of infection [17,66,67,68], heart-infiltrating (IFN-γ-producing) effector T cells drastically reduce the intracardiac load of parasites [17]. On the other hand, chronic myocarditis is worsened as result of unbalanced infiltration of pathogenic subsets (perforin-expressing) of effector CD8 T cells. Notably, we found that R-954 had multiple cardioprotective effects: the daily blockade of B1R (120–160 dpi) attenuated chronic myocarditis, reduced mortality indexes, and ameliorated the conduction abnormalities that are associated with CCC, including QRS (which reflects the duration of the depolarization of both ventricles) and QTc interval (a prolonged QT interval is usually a predictor to increased risk of arrhythmias). Of further interest, in a separate series of experiments conducted with sedated mice, we found that R-954 reversed the increased PR intervals induced by chronic infection (150 dpi, Appendix A). In contrast, the PR interval registered in the absence of sedation was decreased in chronically infected mice (120–160 dpi), suggesting that diazepam contributed to the increased PR interval observed in the sedated group of infected mice [69].

Albeit speculative, it will be interesting to evaluate whether the sudden accumulation of plasma fluids within the intercellular spaces separating heart-conducting fibers might trigger cardiac arrythmia during chronic infection. MRI studies may clarify whether the intracardiac activation of pathogenic subsets of effector CD8 T cells may induce a prolonged myocardial edema via the B1R pathway. For reasons that are unclear, a small number of chronically infected mice were refractory to the anti-fibrotic effects of R-954. Additional studies are required to evaluate whether dysregulated activation of the KKS/DABK/B1R/ACE2 axis might dampen TGF-β-driven cardiac remodeling/fibrosis [52] via the activation of matrix metalloproteinases (MMP2/MMP9) [70,71].

From the perspective of disease prognosis, it will be important to determine whether the infiltration of heart tissues by different subsets of effector CD8^+^ T cells (perforin and/or IFN-γ-producing) results in the upregulated expression of ADAM-17/TACE in the inflamed myocardium. By promoting the proteolytic cleavage and shedding of the ectodomain of ACE2 from the surface of cardiovascular cells, ADAM-17/TACE may indirectly (i) drive the hyperactivation of the pro-inflammatory KKS/B1R pathway and (ii) increase the levels of angiotensin II, indirectly stimulating fibrosis via the activation of angiotensin II receptors [39,40]. For similar reasons, it will be important to evaluate whether dysfunctions of mitochondria energy metabolism in CCC patients [15] translate into a loss of ACE2-dependent regulation of the B1R pathway. If these premises are met, it will be worth considering therapeutic interventions using the combination of two drugs: angiotensin II receptor (AT1) blockers and antagonists of B1R. 

In summary, the results of studies of KKS function in two mice models of *T. cruzi* infection suggest that the activation of the DABK/B1R pathway might fuel heart parasitism during the acute stage of infection. Beyond the impact of KKS activation on *T. cruzi* infectivity, we provided evidence that activation of B1R worsens the severity of chronic myocarditis and fibrosis. The findings that R-954 reduced mortality indexes, inhibited chronic myocarditis, and ameliorated disturbances in heart impulse conduction during the late stage of infection suggest that severity of chronic chagasic cardiomyopathy may be attenuated via pharmacological targeting of the KKS/B1R pathway. 

## Figures and Tables

**Figure 1 jcm-12-02888-f001:**
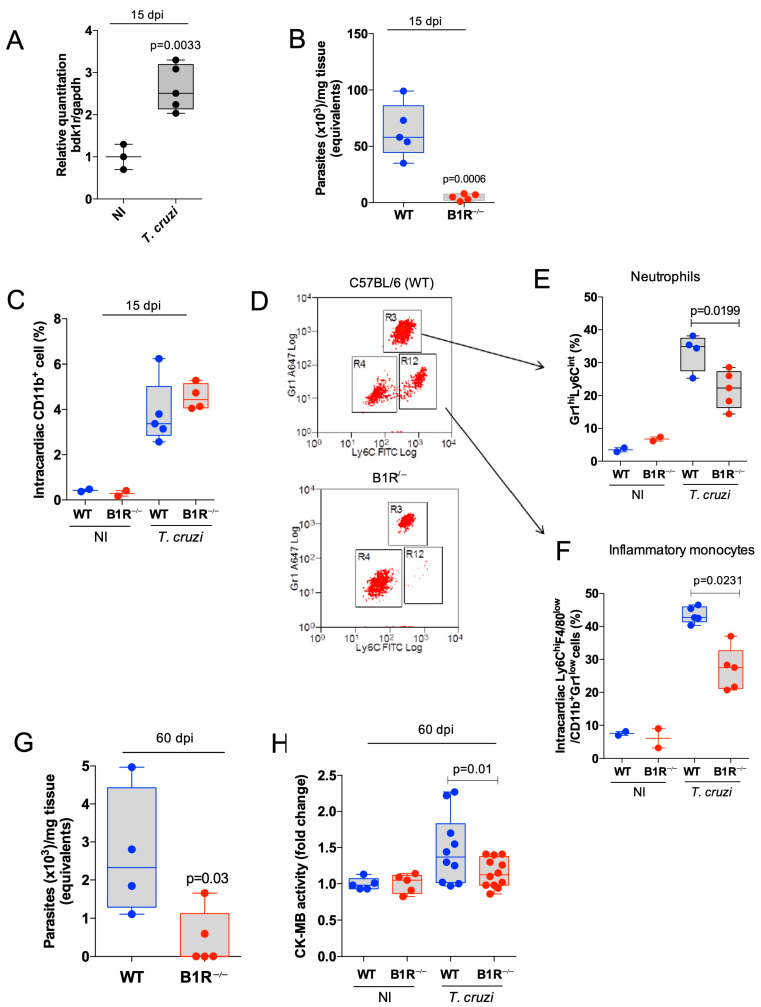
Heart parasitism and severity of cardiac inflammation are attenuated in B1R-deficient mice acutely infected by Dm28c *T. cruzi*. C57BL/6 and B1R^−/−^ mice were infected systemically (i.p.) with Dm28c TCTs (10^6^). At 15 days pi., the intracardiac levels of B1R mRNA (**A**) and *T. cruzi* DNA (**B**) were analyzed by qPCR. Frequencies of CD11b^+^ total cells (**C**) and subsets of inflammatory monocytes (CD11b^+^Gr1^lo^Ly6C^hi^F4/80^lo^) (**E**) and neutrophils (CD11b^+^Gr1^hi^Ly6C^int^) (**F**) in cardiac tissue (15 dpi) by flow cytometry. (**D**) Dot plot representative of gate strategy in CD11b^+^ cells. (**G**,**H**) Extent of heart parasitism (qPCR) (**G**) and serum activity of CK-MB (**H**) measured at 60 dpi. The results are representative of at least three independent experiments; bar shows media ± SD and each symbol represents one mouse (n = 5 mice/group/experiment). In H, results represent the sum of 2 independent experiments. Statistical analysis was performed by *t*-test or one-way analysis of variance (ANOVA) test. *p* values < 0.05 were considered significant. NI, non-infected.

**Figure 2 jcm-12-02888-f002:**
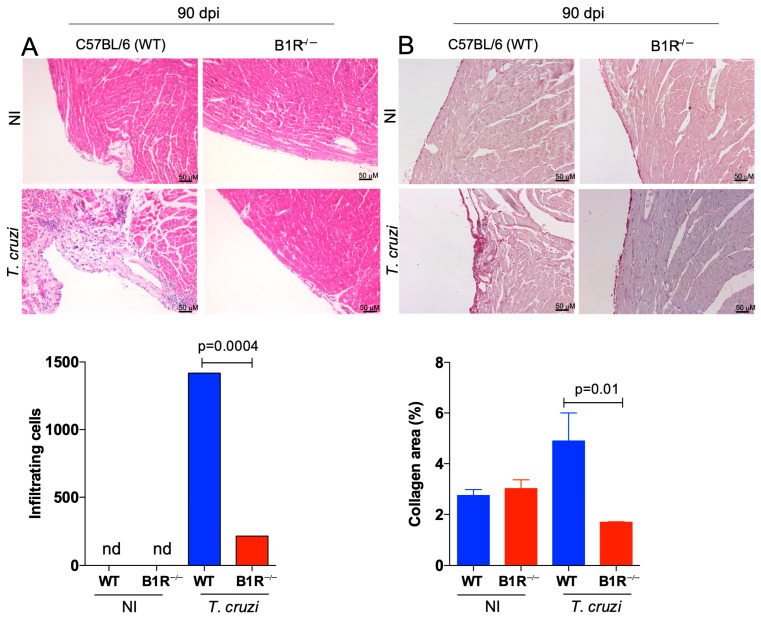
Chronic myocarditis and fibrosis are blunted in *T. cruzi*-infected B1R^−/−^ mice. C57BL/6 (WT) and B1R^−/−^ mice were infected systemically (i.p.) with a lower dose of Dm28c TCTs (1 × 10^3^ TCT). Hearts collected at 90 dpi were fixed in formalin (5%) and paraffin-embedded tissue. Hearts sections (4 μm) were stained by H&E to quantify inflammatory infiltrating cells (**A**) or to measure fibrotic areas (collagen fibers) by Picrosirius Red staining (**B**). Controls were hearts from non-infected mice (received PBS ip.; NI). The results are representative of three independent experiments (n = 7–10 mice/group/experiment). Statistical analysis was performed by one-way analysis of variance (ANOVA) test. *p* values < 0.05 were considered significant. nd, non-detected.

**Figure 3 jcm-12-02888-f003:**
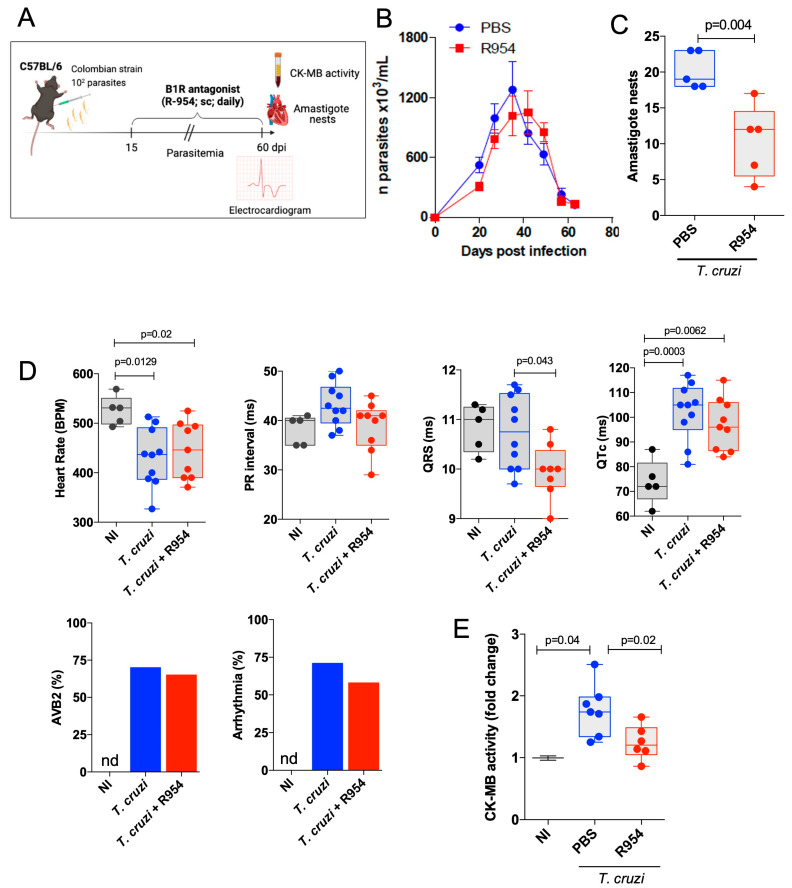
Beneficial effects of early B1R targeting in WT mice systemically infected by the myotropic Colombian *T. cruzi* strain. (**A**) Schematic figure of the infection model and treatment protocol. C57BL/6 (WT) mice were systemically infected (i.p.) with bloodstream Col TCTs (10^2^ parasites). Daily B1R antagonist (R-954) treatment (1.6 mg/Kg; s.c.) started at 15 dpi and lasted up to 60 dpi., whereas the control mice received PBS. (**B**) Parasitemia was followed daily. After 60 dpi, heart tissues were fixed and sections (3 μm) were stained with *T. cruzi*-specific antibody for detection of amastigote nests. Results expressed in 100 fields (**C**). (**D**) Electrocardiogram records obtained at 60 dpi in diazepam sedated mice C57BL/6 mice: heart rate (bpm), PR interval, QRS, QTc, AVB2, and frequency of arrhythmias. (**E**) CK-MB activity was quantified in the serum and the results expressed by fold change. The results are representative of two independent experiments; bar shows media ± SD and each symbol represents one mouse (n = 8–10 mice/group/experiment). Statistical analyses were performed by one-way analysis of variance (ANOVA) test. *p* values < 0.05 were considered significant. nd, non-detected.

**Figure 4 jcm-12-02888-f004:**
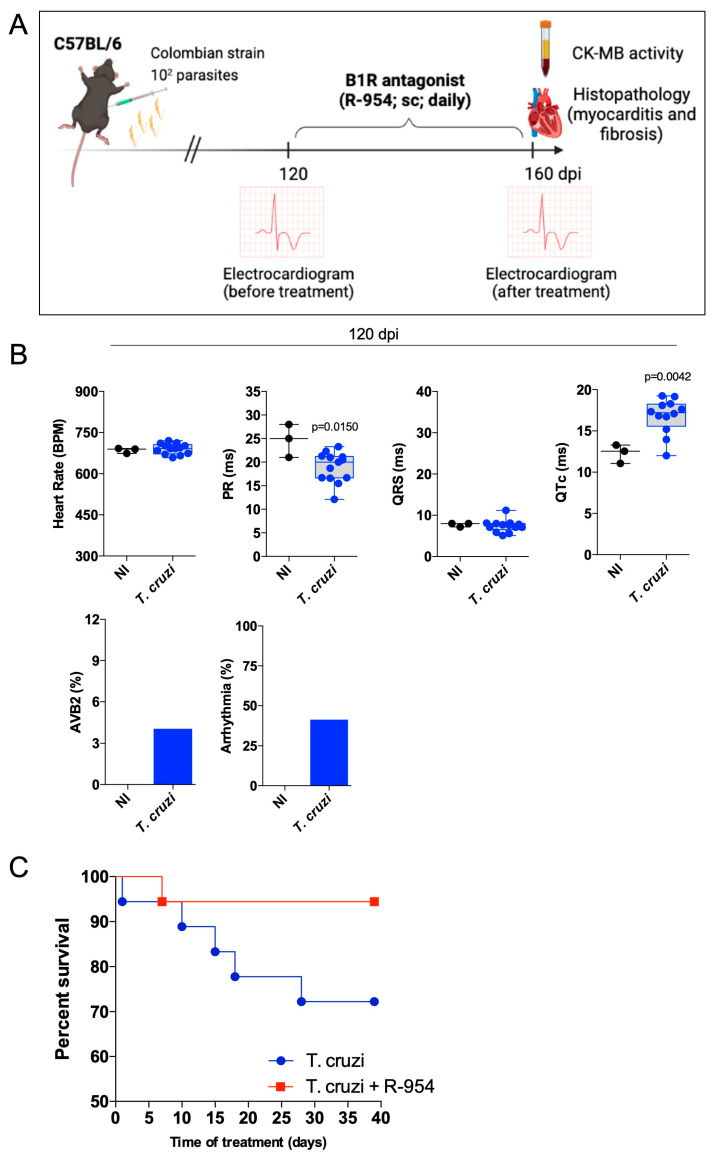
Therapeutic application of B1R antagonist reverses chronic chagasic cardiomyopathy (**A**) Schematic figure of the infection model and treatment protocol. C57BL/6 (WT) mice were systemically infected (i.p.) with Col TCTs (10^2^ parasites). Daily R-954 treatment (1.6 mg/kg—s.c.) started at 120 days pi. and lasted up to 160 days pi., whereas the control mice received PBS. (**B**,**D**) Electrocardiogram registers were performed at 120 ((**B**); before treatment) and 160 dpi ((**D**); after treatment) in awake mice. The parameters were heart rate (bpm), PR interval, QRS, QTc, AVB2, and frequency of arrhythmias. (**C**) Survival rate was followed daily. (**E**–**G**) At 160 dpi, hearts sections were stained by H & E to evaluate the area of cardiac tissue presenting cellular nucleus. The yellow arrows indicated the presence of inflammatory infiltrate. (**F**) or were stained by Picrosirius Red to evaluate fibrosis (collagen area) (**G**). (**H**) CK-MB activity was quantified in the serum and the results expressed by fold change. The results are representative of two independent experiments; the bar shows media ± SD and each symbol represents one mouse (n = 8–10 mice/group/experiment). Statistical analysis was performed by one-way analysis of variance (ANOVA) test. *p* values < 0.05 were considered significant.

## Data Availability

Not applicable.

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
