# Peer review of "Genetic Ablation and Pharmacological Blockade of Bradykinin B1 Receptor Unveiled a Detrimental Role for the Kinin System in Chagas Disease Cardiomyopathy"

_jcm, 2023, doi:10.3390/jcm12082888_

Round 1
Reviewer 1 Report
This paper by Oliveira et al. provides compelling evidence for a detrimental role of the bradykinin B1 receptor (B1R) in Chagas disease cardiomyopathy using two mice models of Trypanosoma cruzi infection and B1R knockout mice. Data suggest that activation of the B1R pathway fuels heart parasitism during the acute stage of infection and worsens severity of chronic myocarditis and fibrosis. Pharmacological blockade of the B1R with R-954 reduced mortality indexes, inhibited chronic myocarditis and ameliorated disturbances in heart impulse conduction during late stage of infection. It is concluded that pharmacological blockade of the proinflammatory B1R pathway might be cardioprotective in acute and chronic Chagas disease.
This is a well conducted and comprehensive study using an experimental approach, which is scientifically sound, and the data support the conclusion. Authors provided a critical analysis of their findings and highlighted potential limitations of the study. This study is clinically relevant as no treatment is currently available for this fatal parasitic disease.
No major flaws are raised by this reviewer who suggests to address minor issues for improvement.
Be careful as the link to access to Supplementary Materials www.mdpi.com/xxx/s1 shows Error 404 - File not found.
Title: change Bradykinin b1 receptor to Bradykinin B1 receptor
Abstract: define (dpi)
Introduction (line 138): des-Arg-lysyl10-BK (DLBK) should read lys-des-Arg9-BK (or des-Arg10-Kallidin). Lysyl is at the N-terminal and not C-terminal.
Material and Methods (para 2.2): It is not clear why the first model of T. cruzi infection is made in male mice while the second model of infection (Colombian T. cruzi) is in female mice. Is the outcome of the infection affected by the sex of the animal?
Para 2.3 (line 198): 5´ CTGTGCATGGCATCAT 3´ probe. Is there redundancy here since the sense and antisense sequences for B1R are given?
Para 2.3 (line 200): 5´CACCAGGGCTGCTT 3´ probe. Again, why this sequence is provided since the sense and antisense sequences for GAPDH are given?
Para 2.3 (line 202): to determinate determine (delete determinate)
Para 2.9: concentrations should be written the same way throughout the paper as mg/ml or µg/ml and not µg.mL-1 (see lines 285, 286, 292, 298).
Para 2.9 (line 293): ACE inhibitor lisinopril will primarily prevent the degradation of kinins and not acts as allosteric modulator of B1R, which is still controversial and not consensual.
Para 2.9 (line 297): B1R antagonist [Leu]8des-Arg9-BK should read [Leu8]des-Arg9-BK
Para 2.13 (lines 340-341): B9858 was a gift from John M. Stewart (University of Alberta, Edmonton, Canada). First, B9858 was not used in this study. Second, John Stewart was living in Denver, Colorado and died several years ago. Are there any preliminary experiments carried out with this antagonist and pooled with data identified as R-954 treated mice? This is quite puzzling!
R954 should also read R-954 throughout.
Figure 2 legend: provide Scale bar in A and B. Line 423: read to determine (instead to determinate).
Line 450: pretreated with Z11777 or K11777 according to Figure S2B?
Figure 3: in A writes correctly Electrocardiogram, In D writes correctly Arrhythmias
Figure 4: in A, B and D, Electrocardiogram and Arrhythmias are not correctly spelled.
Figure 4 C: why the survival rate was not registered for 40 days of treatment but limited to 30 days? Since about 30% of mice died by 160 dpi, how this was taken into account in the statistical analysis of electrical parameters regarding comparison with different n values ?
Author Response
Reviewer #1
Minor concerns:
Starting the point-to-point reply, we wish to thank you for the meticulous correction of the MS. We are sorry that these errors were not spotted in our final revision.
1) Title: change Bradykinin b1 receptor to Bradykinin B1 receptor.
Answer: In the original draft, the title was written in capital letters. It seems that the publisher’s software has automatically converted B into b1 (page 1, line 2).
2) Abstract: define (dpi).
Answer: “Dpi” means “days post infection”. The information was added to the abstract (page1, line 29).
3) Introduction (line 138): des-Arg-lysyl -BK (DLBK) should read lys-des-Arg -BK (or des-Arg -Kallidin). Lysyl is at the N-terminal and not C-terminal.
Answer: Thanks for pointing out this typing error. Since we should have readers that are not familiar with KKS history, des-Arg-Kallidin will not be understood. Let´s stay with lysyl-des-Arg-BK (page3, line 139).
4) Material and Methods (para 2.2): It is not clear why the first model of T. cruzi infection is made in male mice while the second model of infection (Colombian T. cruzi) is in female mice. Is the outcome of the infection affected by the sex of the animal?
Answer:
1. Literature.
Female mice are known to be more resistant to acute infection by Trypanosoma cruzi (Chapman Jr. W.L. et al. J of Parasitology 1975; Vorraro F. et al. Mediators of Inflammation 2014). We included a brief note and the first citation in material and Methods (page 4, line 181, refs 44 and 45).
2. Explanation for the gender choices in studies with B1KO and R-954.
Since WT males are increasingly susceptible to acute infection, they provide a more convenient positive control to compare the extent of heart parasitism (15 dpi) in WT versus B1R-deficient mice. Furthermore, we knew from previous studies (WT vs B2RKO mice-see Monteiro AC et al., 2007) that, at 15 dpi, Dm28c strain of T. cruzi consistently reached the heart tissues of C57BL/6 male mice. To make sure that the Dm28c-infected male mice (susceptible) would last longer-reaching the chronic phase- we simply reduced the inoculum of Dm28c TCT.
The justification for using females (resistant phenotype) in our preclinical studies with R954 is as follows: Since the myotropic Colombian strain is highly virulent, it is preferable to challenge female mice (resistant phenotype) to minimize the mortality during the acute phase and later (see your last query)-during the transition to chronic phase.
.3 (line 198): 5´ CTGTGCATGGCATCAT 3´ probe. Is there redundancy here since the sense and antisense sequences for B1R are given?
Answer: Dr. João Pesquero confirmed that we did not use Taqman. Daniele Andrade has performed these qPCR experiments using SYBR-green fluorescence quantification system. Please note that the correct sequences of the primers for mouse B1R and GAPDH are now informed in the final text.
6) Para 2.3 (line 200): 5´CACCAGGGCTGCTT 3´ probe. Again, why this sequence is provided since the sense and antisense sequences for GAPDH are given?
Answer: Please check our answer to the previous question.
7) Para 2.3 (line 202): to determinate determine (delete determinate).
Answer: This error was corrected (page 4, line 204).
8) Para 2.9: concentrations should be written the same way throughout the paper as mg/ml or μg/ml and not μg.mL-1 (see lines 285, 286, 292, 298).
Answer: Thanks, the description of the concentrations was now standardized.
9) Para 2.9 (line 293): ACE inhibitor lisinopril will primarily prevent the degradation of kinins and not acts as allosteric modulator of B1R, which is still controversial and not consensual.
Answer: True. We removed both the statement and the punctual citation.
10) Para 2.9 (line 297): B1R antagonist [Leu] des-Arg9-BK should read [Leu]des-Arg–BK.
Answer: The error was corrected (page 6, line 298; page 7, line 340).
11) Para 2.13 (lines 340-341): B9858 was a gift from John M. Stewart (University of Alberta, Edmonton, Canada). First, B9858 was not used in this study. Second, John Stewart was living in Denver, Colorado and died several years ago. Are there any preliminary experiments carried out with this antagonist and pooled with data identified as R-954 treated mice? This is quite puzzling! R954 should also read R-954 throughout.
Answer: As you suspected, the supplementary figure S2 of our MS initially contained a series of interesting in vitro results obtained with B9858. Our student reproduced the conditions used 20 years ago in our first study of B1R function in T. cruzi invasion (Todorov et al., 2003). John Stewart collaborated with us, providing B9858 while he was in Denver. The new data (B1R upregulation by conditioned medium collected from Dm28c TCTs) was deleted from the final MS submitted to JCM because they are not relevant to this work. Of course, John Stewart’s generous donation of B9858 was acknowledged in the preliminary draft because my younger colleagues were not aware that this pioneer of the KKS field has left us long ago… As the corresponding author, I apologize for overlooking the content of the acknowledgement section. The inappropriate (puzzling) statement was deleted from the revised MS.
12) Figure 2 legend: provide Scale bar in A and B.
Answer: The scale bar was now included in figure 2.
13) Line 423: read to determine (instead to determinate).
Answer: The error was corrected (page 10, line 424).
14) Line 450: pretreated with Z11777 or K11777 according to Figure S2B?
Answer: “K11777” is correct. Typing error (page 11, line 447).
15) Figure 3: in A writes correctly Electrocardiogram, In D writes correctly Arrhythmias.
Answer: The error was corrected in the text (page 12, lines 464 and 466) and also in the figures.
16) Figure 4: in A, B and D, Electrocardiogram and Arrhythmias are not correctly spelled.
Answer: Gross errors- both were corrected in the final MS (page 16, lines 515 and 517).
17) Figure 4 C: why the survival rate was not registered for 40 days of treatment but limited to 30 days? Since about 30% of mice died by 160 dpi, how this was taken into account in the statistical analysis of electrical parameters regarding comparison with different n values?
Answer: Good points.
First query: Mortality indexes remained stable after 30 and 40 days of R-954 treatment, respectively. To reflect these findings, we had to correct the survival curve, now extended to 40 days of drug treatment.
Second query. As you properly deduced, mortality after 120 dpi was higher in the group of chronically infected mice that were NOT treated with R-954 (positive controls). However, based on a serious of studies performed by our collaborators from Fiocruz/RJ- see Silverio et al, 2012), we predicted that the number of mice in the positive control group (untreated by R-954) would be gradually diminish due to mortality. The experimental design compensated this risk by increasing the number of mice of the positive control (untreated) group. We used 10 to 8 mice, depending on the experiment. A rate of 30% of mortality was confirmed, leaving us with 6 or 7 survivors in the positive control group at the end-point of the study. Although the number of R-954-treated mice were smaller, the survival rates were higher, making the experiments amenable to statistical analyzes. An interesting point came to mind as we answered your query: since the mortality is higher in the group of untreated mice, the cardioprotective effects of R-954 could have been far more pronounced if the ECG measurements had been made earlier, before death of the sickened mice.

Reviewer 2 Report
Major and minor concerns:
1) It would be desirable to include a western-blot showing the lack of expression of the B1R in knock-our mice and parasitemia in both animal models.
2) The lack of immune response is due to the low parasitemia or the result of any other mechanims related to the absence of the gene?
3) In Figure 1, the number of animals per experiment is different in which part, and not 5 as written in the legend
4) It would be desirable to include the data corresponding to NI animals for parameters shown in Supp Fig 1, part A, B and C
5) In 388, other cells than antigen-experienced T cells could secrete IFN-gamma detected by ELISA. I suggest to change this paragraph or explain better because authors could have done additional experiments to support this statement
6) In the same line as 5), in line 396, the statement is too extensively when the functional activity is only limited to a peptide-specific T cell.
7) Line 440, all this paragraph is an speculation and is also mentioned in the Discussion. None experiments have been done to propose the shedding of extracellular vesicles containing tGPI-mucins. Also, other factors could be involved in the different capacity of T. cruzi to invade and infect cells.
8) Is diazepam an analgesic that does not alter heart parameters in mice? What was the goal behind carrying out both models?
9) Line 610, what is the meaning of this paragraph: "Strictly analyzed from the perspective of experimental design, the first part of our study suffered from the fact that global ablation of b2r or b1r genes tend to overexpress B1R or B2R in a compensatory manner", happened this in your model? could you show it? And if so, how can the authors explain the results based only on the lack of B1R?
10) Could the authors explain why the effect of B1R blocker is effective in the chronic but not in the acute phase of infection?
11) Change some T. cruzi in italic
12) Some punctiation marks are missing
Author Response
Reviewer #2 - Please see the attachment to see figures.
Major and minor concerns:
1) It would be desirable to include a western-blot showing the lack of expression of the B1R in knock-our mice and parasitemia in both animal models.
Answer: Past efforts to use commercial antibody to mouse B1R (Santa Cruz) were frustrating: we were not convinced of its alleged specificity. Perhaps aware of this old problem, the company is now selling a monoclonal antibody. We rushed to import this new antibody, but shipment to Brazil is always sluggish. It will be impossible to meet this desirable goal in the short term. However, in a couple of months, we should know whether this tool is finally available for the scientific community.
2) The lack of immune response is due to the low parasitemia or the result of any other mechanisms related to the absence of the gene?
Answer: “lack of immune response…”. There is NO “lack of immune response”, either in infected WT or B1R-/- mice. On the contrary, anti-parasite immunity (15 dpi) is vigorous (heart or spleen) in both WT and B1R-deficient lines, as we can see in the Figure S1 in the original manuscript. These results imply that absence of B1R does not influence the intralymphoid generation nor the intracardiac recruitment of immunoprotective T cells.
Regarding the participation of other mechanisms: In the original manuscript we showed that the intracardiac frequencies of GzB-and IFN-g-expressing CD8 T cells and IFN-g+CD4 T cells are preserved in the B1R-deficient mice. Beyond these findings, we know that the frequency of intracardiac GzB+CD4+ cells, a cytotoxic subset involved in anti-parasite immunity (Barbosa et al. Cytotoxic CD4+ T cells driven by T-cell intrinsic IL-18R/MyD88 signaling predominantly infiltrate Trypanosoma cruzi-infected hearts, ELife 2022) was likewise preserved in the B1R-/- mice (Data shown in the below graph, A). Moreover, frequencies of NK1.1+ cells (B) and TCR gd+ T cells (C) were also similar in the absence of B1R. We did not add these additional data to the revised MS because this is not the main point of the paper and figure S1 would containing excessive information. If you and the editors decided otherwise, we will incorporate these graphs in the supplementary Figure 1.
Frequencies of intracardiac CD4+GzB+ (A), NK1.1+ (B) and CD3+gd+ (C) cells in the control (NI) and T. cruzi infected C57BL/6 and B1R-/- mice. Flow cytometry was performed at 15 days post infection.
3) In Figure 1, the number of animals per experiment is different in which part, and not 5 as written in the legend.
Answer: Thanks for pointing out this flaw. The graph showing CK-MB activity (Figure 1G) represents the sum of 2 independent experiments (total n = 10 mice). This information was corrected in the figure legend (page 9, line 367).
4) It would be desirable to include the data corresponding to NI animals for parameters shown in Supp Fig 1, part A, B and C.
Answer: Irrespective of antigen specificity, the frequency of heart infiltrating CD4 and CD8 T cells is extremely low in non-infected mice (please check the graph below). Please note that this functional analysis was performed in a previous article published by one of us (Oliveira, A.C. et al. Crucial role for T cell-intrinsic IL-18R-MyD88 signaling in cognate immune response to intracellular parasite infection, Elife 2017).
Frequencies of intracardiac CD4+ (A) and CD8+ (B) T cells in the control (NI) and T. cruzi infected C57BL/6 and B1R-/- mice. Flow cytometry was performed at 15 days post infection.
5) In 388, other cells than antigen-experienced T cells could secrete IFN-gamma detected by ELISA. I suggest to change this paragraph or explain better because authors could have done additional experiments to support this statement.
Answer: The paragraph was altered: “Using ELISA, we then compared the IFN-γ secretion by splenocytes from WT and B1R knockout infected mice exposed to soluble epimastigote extracts or the immunoprotective H-2Kb-restricted TsKb20 peptide. In both cases the response was not enhanced in infected B1R-/- mice (Figure S1G)”.
6) In the same line as 5), in line 396, the statement is too extensively when the functional activity is only limited to a peptide-specific T cell.
Answer: T. cruzi is thought to escape from adaptive immunity because it expresses thousands of polymorphic genes. The number of MHC-class I-restricted epitopes that can be potentially recognized by effector T cells is huge. T. cruzi trans-sialidase (Vasconcelos et al. Protective Immunity Against Trypanosoma cruzi Infection in a Highly Susceptible Mouse Strain After Vaccination with Genes Encoding the Amastigote Surface Protein-2 and Trans-Sialidase, HumGeneTher 2004) is the source of Tskb20, one of the most extensively characterized immunodominant peptide analyzed in mice models of Chagas disease. This is the reason why we compared WT and B1R-/- immune responses against this MCH I-restricted peptide.
7) Line 440, all this paragraph is a speculation and is also mentioned in the Discussion. None experiments have been done to propose the shedding of extracellular vesicles containing tGPI-mucins. Also, other factors could be involved in the different capacity of T. cruzi to invade and infect cells.
Answer: You are right. We deleted the paragraph and citation.
8) Is diazepam an analgesic that does not alter heart parameters in mice? What was the goal behind carrying out both models?
Answer: The model of CCC established with the myotropic Colombian strain was chosen for ECG measurements because it was extensively characterized by our collaborators (JLV) at Fiocruz. We chose to perform the first set of ECG measurements using diazepam-treated animals because this was routinely done at their lab-we needed a gold standard. Having obtained promising results with R-954, we reasoned that the second set of experiments conducted with R-954 should be preferably conducted with analgesic-free mice because (i) early studies in dogs showed evidence that ECG registers are influenced by diazepam (Nakajima et al 1979; this reference was cited in our text, page 19, line 695); (ii) ECGs in humans are performed in the absence of analgesics. Although the values of the conduction parameters were different, R-954 ameliorated the dysfunction in the absence or presence of analgesics.
9) Line 610, what is the meaning of this paragraph: "Strictly analyzed from the perspective of experimental design, the first part of our study suffered from the fact that global ablation of b2r or b1r genes tend to overexpress B1R or B2R in a compensatory manner", happened this in your model? could you show it? And if so, how can the authors explain the results based only on the lack of B1R?
Answer: The questions that you have raised are interesting from the standpoint of basic immunology (KKS function in innate and adaptive immunity). In theory, we could try to reassess the profile of anti-parasite immunity in B1KO mice following systemic treatment with icatibant, administered at different time-windows. Since the main interest of the current work was to evaluate the impact of B1R gene deletion or blockade in the outcome of chagasic cardiomyopathy, the unresolved issues (B2R versus B1R function in innate/adaptive immunity) must be left for future studies.
10) Could the authors explain why the effect of B1R blocker is effective in the chronic but not in the acute phase of infection?
Answer: excellent question. The blood parasitemia in the acute phase is dominantly reflecting the extent of parasite burden of splenic macrophages. Since the liver and spleen are irrigated by sinusoids (open circulation), antibodies and plasma proteins diffuse freely through fenestrated endothelium. Under these circumstances, the fate of T. cruzi inside splenic macrophages (replication and morphogenesis) is largely determined by the efficiency of phagocytic uptake and activation state (which limits intracellular outgrowth of parasites). Targeting of B1R (sinusoids or macrophages) may Snot significantly disable both functions. This might explain why blood parasitemia is not reduced by R-954. A very different scenario may prevail in tissues irrigated by non-fenestrated capillaries, such as the heart and gastro-intestinal tract. By the time that T. cruzi reaches the heart (15 dpi), it is unlikely that cardiac macrophages might severe as niches for parasite outgrowth. They may be already activated by TNF-a and IFN-G-secreted by infiltrating T cells. Taking advantage of their biological versatility, the trypomastigotes may have increased survival chances by preferentially infecting cardiovascular cells, fibroblasts, or even recruited stem cells that express a broad range of GPCRs. Kinins and endothelins may be just examples of short-lived peptides that propagate the inflammatory wave while fueling parasite uptake via GPCR signaling. Using an intracardiac model of T. cruzi infection in naïve mice (Nascimento et al., 2017), we provided evidences that heart parasitism can be fueled via extravascular activation of MC/KKS pathway. In the present work, it is conceivable that R-954 has stabilized the endothelial barrier. By reducing the extent of proteolytic generation of kinins and other signaling peptides in the inflamed heart tissues, the B1R blocker may have limited extent of invasion of heart cells. Since the article is already dense, we chose to delete this complex explanation (discussed in our articles) from the main text.
11) Change some T. cruzi in italic.
Answer: Thank you. The correction was done.
12) Some punctiation marks are missing.
Answer: Thank you. The correction was done.

Round 2
Reviewer 2 Report
I agree with authors that Discussion is extensively enough to add any other speculation. However it would be interestingly to discuss the difference observed between acute and chronic infection when mice were treated with the B1R blocker by summaring some paragraph of this section.
In addition, I suggest to discard the data with diazepam, which did not improve the manuscript and is not relevant.
Some T. cruzi are not in italic.
Author Response
Reviewer #2
Minor concerns:
Reviewer: I agree with authors that Discussion is extensively enough to add any other speculation. However, it would be interestingly to discuss the difference observed between acute and chronic infection when mice were treated with the B1R blocker by summaring some paragraph of this section.
Author response: As requested, we inserted a small paragraph (page 18, lines 655-664), offering an interpretation for the discrepant effects of R-954 in the acute phase (reduced heart parasitism versus no effect on blood parasitemia).
Concerning the cardioprotective effects of R-954 during chronic infection, our interpretation (hypothesis) was outlined in the original MS. Please read lines 714-715 of the current MS. Note that we have highlighted “during chronic infection” to make sure that readers will follow our reasoning.
Reviewer: In addition, I suggest to discard the data with diazepam, which did not improve the manuscript and is not relevant.
Author response: We ask you to kindly reconsider this request- for the following reasons.
i. ECG measurements in diazepam-treated mice were extensively used in several (excellent) studies published by our collaborators at Fiocruz. It is technically hard to make ECG measurements with awake mice. Preserving the complete set of data will make justice to the efforts made by several members of our labs- each of these experiments lasted 5-6 months (each) and were partially conducted with sacrifice during the COVID-19 pandemics.
ii. Although the numeric values of conduction responses differed in infected mice treated or not with diazepam, it was very reassuring to observe that R-954 had cardioprotective effects in both procedures. Note that in absence of diazepam data, the impact of R-954 in heart conduction would be limited to N=1, which is not necessary, nor desirable-for obvious reasons.
iii. As explained in our first reply, we justified in the text (original MS) why the ECG measurements in our second experiment (N=2) were performed without analgesics.
iv. We chose to OPEN THE REVIEW process; hence readers will have access to this debate.
Reviewer: Some T. cruzi are not in italic.
Author response: The correction was done. In a few places, additional errors were spotted and corrected. The text was gradually improved. Thanks.